# Detecting Montane Flowering Phenology with CubeSat Imagery

**Aji John** [1,*] **, Justin Ong** [2] **, Elli J. Theobald** [1] **, Julian D. Olden** [3] **, Amanda Tan** [4] **and Janneke HilleRisLambers** [1,5]

1   Department of Biology, University of Washington, Seattle, WA 98195, USA; ellij@uw.edu (E.J.T.); jhrl@uw.edu (J.H.)
2   Paul G. Allen School of Computer Science and Engineering, University of Washington, Seattle, WA 98195, USA; cjto2000@uw.edu
3   School of Aquatic and Fishery Sciences, University of Washington, Seattle, WA 98195, USA; olden@uw.edu
4   eScience Institute, University of Washington, Seattle, WA 98195, USA; amandach@uw.edu
5   Institute of Integrative Biology, ETH Zurich, 8092 Zurich, Switzerland
*   Correspondence: ajijohn@uw.edu

**Abstract:** Shifts in wildflower phenology in response to climate change are well documented in the scientific literature. The majority of studies have revealed phenological shifts using in-situ observations, some aided by citizen science efforts (e.g., National Phenology Network). Such investigations have been instrumental in quantifying phenological shifts but are challenged by the fact that limited resources often make it difficult to gather observations over large spatial scales and long-time frames. However, recent advances in finer scale satellite imagery may provide new opportunities to detect changes in phenology. These approaches have documented plot level changes in vegetation characteristics and leafing phenology, but it remains unclear whether they can also detect flowering in natural environments. Here, we test whether fine-resolution imagery (<10 m) can detect flowering and whether combining multiple sources of imagery improves the detection process. Examining alpine wildflowers at Mt. Rainier National Park (MORA), we found that high-resolution Random Forest (RF) classification from 3-m resolution PlanetScope (from Planet Labs) imagery was able to delineate the flowering season captured by ground-based phenological surveys with an accuracy of 70% (Cohen's kappa = 0.25). We then combined PlanetScope data with coarser resolution but higher quality imagery from Sentinel and Landsat satellites (10-m Sentinel and 30-m Landsat), resulting in higher accuracy predictions (accuracy = 77%, Cohen's kappa = 0.39). The model was also able to identify the timing of peak flowering in a particularly warm year (2015), despite being calibrated on normal climate years. Our results suggest PlanetScope imagery holds utility in global change ecology where temporal frequency is important. Additionally, we suggest that combining imagery may provide a new approach to cross-calibrate sensors to account for radiometric irregularity inherent in fine resolution PlanetScope imagery. The development of this approach for wildflower phenology predictions provides new possibilities to monitor climate change effects on flowering communities at broader spatiotemporal scales. In protected and tourist areas where the flowering season draws large numbers of visitors, such as Mt. Rainier National Park, the modeling framework presented here could be a useful tool to manage and prioritize park resources.

**Keywords:** phenology; alpine wildflowers; CubeSat; Landsat; Sentinel-2; PlanetScope; random forest; SWEEP

## 1. Introduction

Shifts in seasonal timing of biological events in plants, such as germination, flowering, and fruiting, in response to climate change have been observed across numerous species [1,2]. For example, many species demonstrate earlier onset of development and advances in other key life-history events as the climate warms [3]. Alpine wildflowers are considered good indicators of climatic change, as their phenology is highly sensitive to spring and summer temperatures and the timing of snowmelt [4]. Such shifts in the timing of flowering of Alpine wildflowers is concerning, as it could disrupt interactions between Alpine wildflowers and other members of the community, such as pollinators [5]. Understanding and anticipating the impacts of climate change on flowering is therefore paramount to help inform conservation efforts to ensure the preservation of wildflower meadows in the future [6].

Mounting evidence from field studies on wildflower phenology compiled by scientists and volunteer networks document shifts in the timing of flowering [7–9] that seem linked to climate warming at those locations [10–12]. However, whether it is possible to extrapolate information gained at smaller spatial scales to larger spatial scales remains uncertain [13]. Quite simply, monitoring flowering phenology across broader spatial and temporal domains is challenged by limited monetary and human resources. The ability to detect flowering phenology via remote sensing, if possible, therefore has the potential to aid in the quantification of fine-scale flowering phenology at large spatiotemporal scales. Hyperspectral and multispectral imagery has proven to be useful when analyzing vegetation type and leaf green-up [14] in crop phenological studies [15–17] but, to the best of our knowledge, has not been used to assess other plant phenological stages in natural systems.

Recent developments of small satellites, so-called CubeSats, have introduced new possibilities to monitor land cover at fine spatial and temporal scales. CubeSats have the advantage of high spatial resolution (in comparison to more traditional satellite imagery) but are limited to few spectral bands and usually have a small form-factor (dimensions in the order of 10 cm by 10 cm by 30 cm). The low costs associated with their deployment have resulted in several companies using CubeSat technology to provide high spatial and temporal coverage of the earth. One such provider is Planet Labs, Inc. [18], which uses over 150+ (as of 2019) CubeSats to image the entire land surface of the Earth at a daily time interval and 3–5 m resolution. This imagery provides ecologists with the exciting opportunity to track a wide array of natural processes that occur at fine spatial scales. Radiometric quality remains an issue of this data [19]; however, this shortcoming may be improved by cross-calibration using highly calibrated coarser resolution imagery, like Landsat and Sentinel [20].

Multispectral bands have long been used to detect plant physiological properties, and CubeSats offer the possibility of doing so at finer spatial resolutions than other commonly used satellite imagery. Multispectral products are produced by examining the reflectance spectra that are captured in the corresponding electromagnetic spectrum and are often used in calculating band ratios [20]. For example, a commonly used band ratio (or radiometric index) that is used to measure vegetation state is NDVI (Normalized Difference Vegetation Index—ratio of red and near infrared bands), which has been found successful when applied to crop type estimation, e.g., in maize, soybean, and wheat, because of the inherent uniformity in the vegetation [21]. In addition, it has been shown that yellow flowers can decrease NDVI values in alpine meadows [22], and a cut-off of 0.4 NDVI has been reported as indicative of full flowering in a study involving sunflower crops [23].

For this paper, we determined whether multi-spectral imagery obtained from Planet Labs, Inc., could help to quantify peak flowering in sub-alpine meadows. We did so by combining satellite imagery with an existing long-term on-the-ground datasets. The long-term datasets identify the growing season phenology of wildflower meadows during a period of five years along an elevation gradient at Mt. Rainier National Park in Washington, USA. We evaluated whether phenological stages are observable via the spectral bands of PlanetScope (a 3-m, 4-band multispectral image product from Planet Labs) by using reflectance in the red and near infrared (NIR) bands. Additionally, we evaluated whether PlanetScope-derived NDVI can be used to delineate flowering. We also combined finer resolution imagery from Planet with coarser resolution imagery from Sentinel (10 m) and Landsat

(30 m) to assess whether more accurate phenological detection is possible when combining high these two types of imagery. Our objective involved the two following questions. First, can peak flowering be detected via either PlanetScope NIR and red bands or a normalized measure, like NDVI? Second, is detectability of flowering improved when fine-resolution PlanetScope is combined with coarser resolution imagery (with higher quality images), like Landsat and Sentinel?

## 2. Materials and Methods

### 2.1. Study Site

Study sites are located in Mt. Rainier National Park (46.8529° N, 121.7604° W; summit elevation 4392 m), part of a stratovolcanic mountain range in the Cascade Range of Washington State, USA. The regional climate is maritime with dry summers and wet winters. The vegetation is dominated by coniferous forests at lower elevations (<1450 m), and sub-alpine wildflower meadows at mid-elevations (1450–1900 m). We restricted our analyses to meadows spanning an elevational gradient, where phenology has been quantified in a long-term study from 2010–2015 (Theobald et al., 2017). Spatial polygons encompassing meadows were manually drawn to include all sampling locations in this study. These meadows are adjacent to a trail where additional long-term monitoring of flowering phenology is ongoing (2013–present) through a citizen science project called MeadoWatch (http://www.meadowatch.org/), providing additional information on flowering phenology after 2015. The selected meadow sites were on the south slope of Mt. Rainier, along Mazama Ridge, approximately 100 m apart (Table 1 and Figure 1).

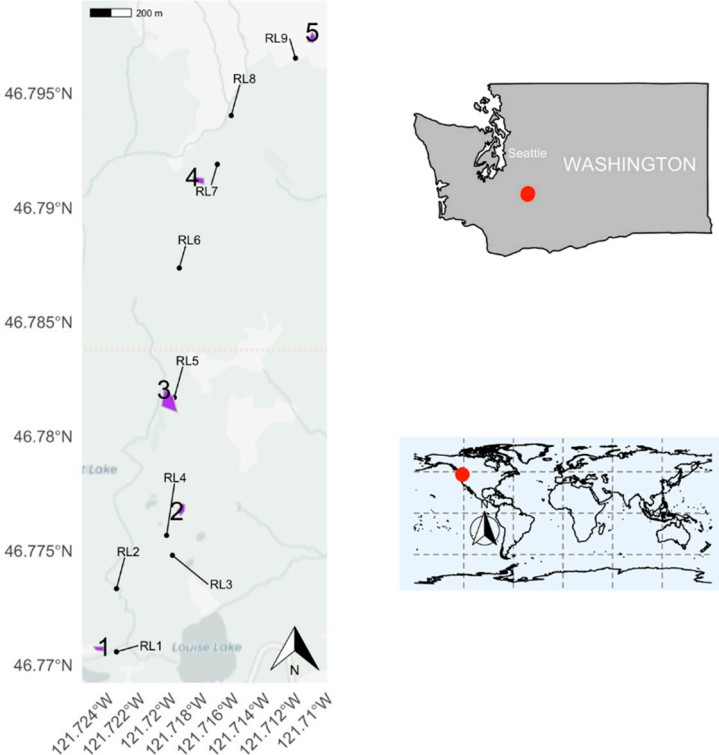

**Figure 1.** Meadow study sites at Mt. Rainier National Park; five sub-alpine meadows colored in purple are approximately 100 m apart in elevation and cover an elevational gradient from 1490 m to 1900 m. Polygons are the sites from the study Theobald et al. (2017), and the dots annotated with RL1-9 are plots from MeadoWatch.

**Table 1.** Elevation and area of meadow sites.

| Meadow | Elevation (m) | Area (m$^2$) |
|---|---|---|
| 1 | 1487 | 877 |
| 2 | 1595 | 1328 |
| 3 | 1678 | 4694 |
| 4 | 1779 | 1357 |
| 5 | 1894 | 1136 |

*2.2. Remote Sensing Data*

Remote-sensed imagery were collected from Landsat, Sentinel, and Planet Labs. PlanetScope imagery from Planet Labs became available only after 2016, so we extracted two years of PlanetScope imagery (2017–2018), 6 years of imagery from Landsat (2013–2018), and 2 years of imagery from Sentinel, which was initiated in 2015 (2017–2018). We excluded data from 2016 because of quality issues with Sentinel imagery (see Appendix D for imagery acquisition and resolution). Thousands (n = 2893) of images of meadows were analyzed over 5 years of study across providers Landsat 8, Sentinel-2, and Planet. For all three providers, we extracted summary reflectance values across red, green, blue, and NIR bands for each meadow area polygon (see Appendix C for band wavelengths). Additionally, we calculated summary reflectance in short-wave infrared (SWIR) and green chromatic coordinate ($g_{cc}$) for Sentinel imagery for each meadow. We briefly describe the three providers below.

The Landsat 8 OLI (Operational Land Imager) platform offers 30-m data with high-quality spectral calibration. The Landsat-8 satellite is one of a series that were launched by the National Aeronautics and Space Administration (NASA)/U.S. Geological Survey (USGS). Landsat Collection 1 Level-1 data products are used in the Landsat analysis (here onwards referred as L8).

Sentinel-2 is owned by the European Space Agency (ESA) and offers 10-m data for red, green, blue, and NIR bands. Officially called the Copernicus Sentinel-2 mission, Sentinel-2 consists of two polar-orbiting satellites in the same sun-synchronous orbit, phased at 180 degrees to each other (Sentinel-2, 2019). Sentinel-2 Level-1B products are used in the Sentinel workflow (here onwards referred as S2-1B).

Planet Labs PlanetScope constellation consists of 120+ CubeSats which orbit in two near-polar, sun-synchronous orbits of ~8° and ~98° inclination angle at an altitude of roughly 475 km. The CubeSats acquire both visible (RGB) and NIR data with 12-bit radiometric resolution that images the entire land surface on Earth daily. Specifically, we used PlanetScope item PSScene4Band (type *analytic_sr*) in the Planet workflow (here onwards referred as PS).

*2.3. Training and Validation Data: Peak Flowering from on-the Ground Observations*

To associate the satellite images to peak flowering, we developed a method to determine peak flowering window by elevation: first, we deduced flowering windows from field-based MeadoWatch observations (collected on the trail adjacent to meadow polygon) observations for 2017 and 2018 when PS, L8, and S2-1B imagery was available. Then for validation, we determined flowering time for 2013–2015 from observations from MeadoWatch and Theobald et al. (2017) when only L8 imagery was available. Example kernel density curves of observed flowering of top 10 recorded species for 6 years from MeadoWatch (2013–2018) are presented in Figure 2; 2015 showed a considerable shift in peak flowering day due to unprecedented warm temperatures (in contiguous United States average temperature was 2.4 °F above the 20th century average). See Appendix D for imagery and study period details.

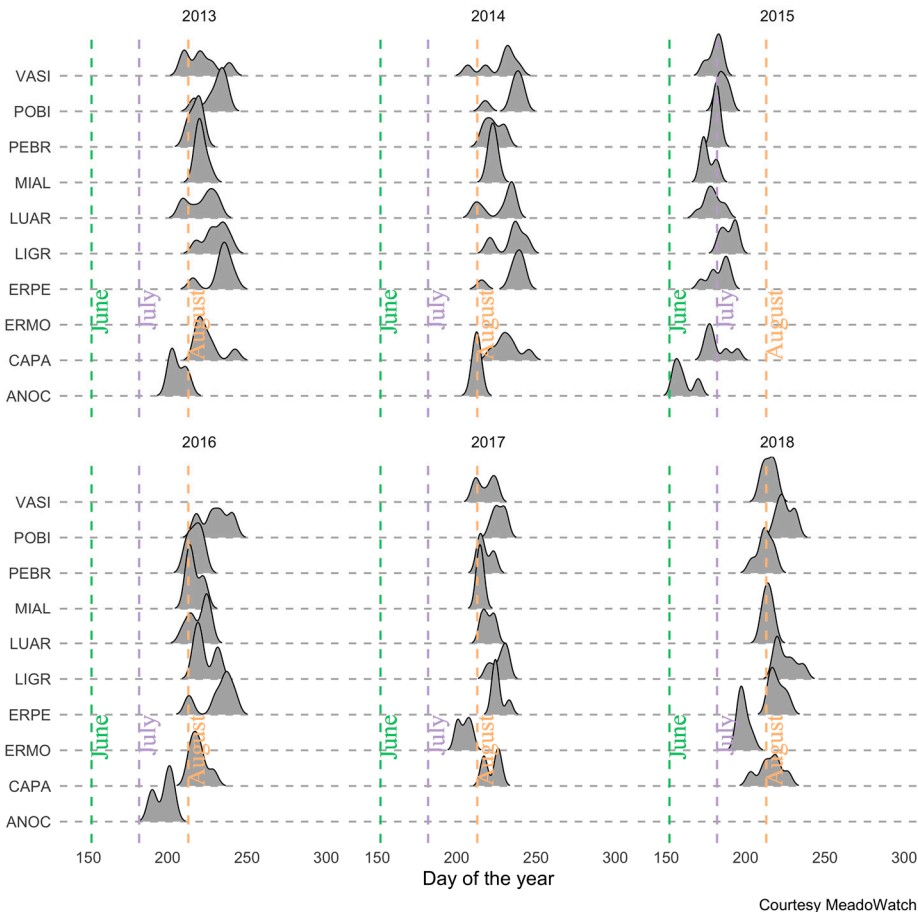

Courtesy MeadoWatch

**Figure 2.** In-situ observed flowering of 10 of the most abundant meadow species across a 6-year period (2016 shown for completeness). The dotted lines indicate start of the month. Species include *Valeriana sitchensis* (VASI), *Polygonum bistortoides* (POBI), *Pedicularis bracteosa* (PEBR), *Microceris alpestris* (MIAL), *Lupinus arcticus* (LUAR), *Ligusticum grayi* (LIGR), *Erigeron peregrinus* (ERPE), *Erythronium montanum* (ERMO), *Castilleja parviflora* (CAPA), *Anemone occidentalis* (ANOC).

Flowering windows were estimated using flowering observations by the MeadoWatch program and those reported in Theobald et al. (2017). In both the cases, observer(s) recorded the species, the date (day of the year) and flowering status ('yes' or 'no') in multiple plots (plots measured 1 m × 1 m in Theobald et al. 2017 and were estimated at ~2 m × 1 m in MeadoWatch). The main difference between these datasets is that multiple 1 m × 1 m plots were sampled within one of five meadows sites by Theobald et al. (2017), whereas sampling occurred at nine single MeadoWatch plots, along a prominent hiking trail. MeadoWatch plots were close to the five Theobald sites, and spanned the same elevation gradient. Plots within the Theobald et al. (2017) sites were combined to delineate an area (purple polygons in Figure 1) to fetch the satellite imagery. For the purposes of this study, the MeadoWatch plots were grouped by elevation using the shortest distance to nearby Theobald site (Figure 1). Once grouped by elevation, the observations were then separated by year to calculate mean and standard deviation (SD) of peak flowering day (Figure 3A). The days in one ±SD of the mean was considered to be peak flowering window for that year and elevation (Figure 3B). Our determination of flowering windows assumes that all 10 focal species are found in all plots in all years. This is not a valid assumption but making it does not bias our conclusions as we are making inference at the community-level. In other words, we are determining peak flower as the date at which there is the highest probability of seeing any species in flower and are using the 10 most prominent and abundant species to make this assessment.

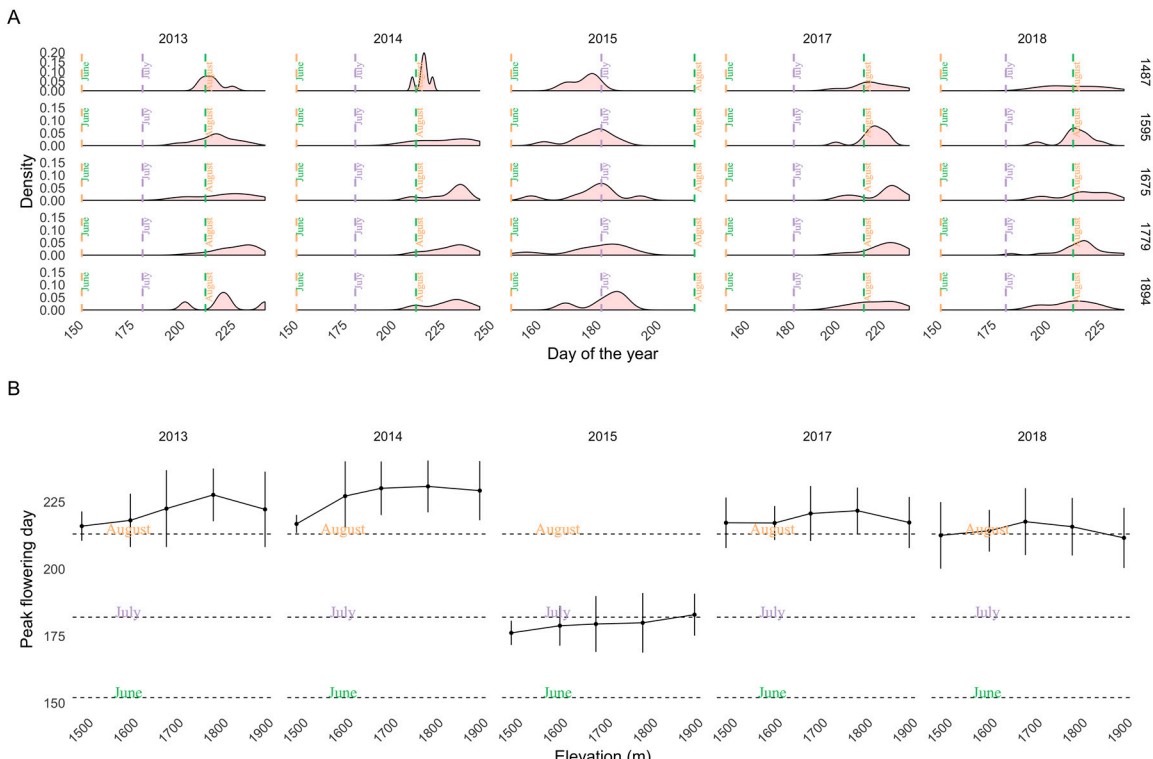

**Figure 3.** Peak flowering windows using MeadoWatch observations. (**A**) Kernel density of flowering observations by elevation (in meters) and year. (**B**) Estimated flowering windows calculated using mean flowering day and ±1 SD; error bars indicate length of the flowering window with the dot signifying mean peak flowering day. Note the shift in flowering phenology in 2015, a historically warm year.

## 2.4. Satellite Data Processing

All processing of satellite imagery data was performed using SWEEP [24], a workflow management platform for distributed execution in cloud infrastructures. SWEEP workflows for acquiring and analyzing imagery were developed for PS, L8, and S2-1B and are briefly outlined in the Appendix B.

## 2.5. Analysis

### 2.5.1. Importance of Spectral Bands in Flowering

To assess the relevance of the NIR and red spectral bands to flowering phenology, we employed Principal Component Analysis (PCA) to summarize dominant patterns of variation in the spectral band data and relate it back to flowering phenology. PCA, a multivariate ordination approach, is a preferable way to account for correlations among spectral bands [25,26]. We hypothesized that a shift in reflectance signature in NIR and red bands would be observed in known flowering months and would differ from the signature when meadows green-up (early summer, after snowmelt) or when the meadows are covered in snow (early summer and spring). Therefore, for each satellite image we extracted the minimum, maximum and average pixel value by spectral band for each meadow. We created a spectral matrix where rows (or objects) corresponded to satellite images (from PS), and columns (descriptors) included summary measures (minimum, maximum, and mean) of all the bands and NDVI (calculated using the corresponding red and NIR band). PCA was conducted using PS imagery only because a fine resolution of 3 m has more spectral representation for an average meadow (typically 30 m by 30 m) and is more likely to capture the spectral variability in a meadow, i.e., 9-pixel values for an average meadow. We used PCA in part because it could potentially help identify the reflectance bands that are related to

months when flowering is typically observed. Hence, we characterized the relationship between the flowering meadows and spectral reflectance to understand which band captures flowering.

### 2.5.2. Flowering Prediction Using Random Forest (RF)

We used a Random Forest (RF) classifier [27] to predict the occurrence of flowering (binary yes or no) as a function of the spectral variables for years 2013–2015. Random Forest (RF) classifiers are a model-averaging or ensemble-based approach in which multiple classification tree models are built using random subsets of the data and predictor variables. This approach uses a recursive partitioning algorithm to repeatedly partition the data set according to the predictor variables into a nested series of mutually exclusive groups, each as homogeneous as possible with respect to the response variable. It requires fewer parameters to be fit, has been shown to be less biased than other machine learning methods, and is known to be effective in classifying vegetation in remote sensing applications [28,29]. We used Cohen's kappa statistic to quantify model performance (comparing expected vs. observed error), and Gini importance to obtain the predictive contributions of the spectral features in the Random Forest (RF) classifier. We picked the threshold for determining the predicted probability of flowering by evaluating the receiver operating curve (ROC) and the distribution of true positive rate (sensitivity) and true negative rate (specificity) with respect to threshold.

We compared how finer resolution imagery compared to the combined resolution imagery by comparing their Random Forest (RF) performance metrics. We qualitatively assessed peak flowering using L8 imagery from the years 2013–2015 with two surveys; in-situ observed flowering survey by Theobald et al., 2017 and MeadoWatch.

All the statistical analyses were performed in R (R Development Core Team 2008).

## 3. Results

### 3.1. Importance of Spectral Bands in Flowering

The PS data showed that reflectance in the NIR band is comparatively higher in July and August, with the other bands showing a high degree of correlation (Figure 4C,D). There is a steady decrease in reflectance from June to July during snowmelt, and then a rapid decrease in reflectance during July and August, aligned with the timing of green-up and flowering.

Strong patterns are evident in red and NIR with respect to temporal patterns in observed flowering. Reflectance in green band is lower when compared to other bands until early July, and then close to zero reflectance in visible bands (i.e., red, green and blue; also means strong absorption). Increasing reflectance in the blue band reflects the snowmelt period until early June, followed by months characterized by decreasing reflectance that indicate greening-up of vegetation. Green-up causes a steady decrease in reflectance in all bands until an inflection point around late August, evident in 2017 and 2018 (Figure 4C,D). Higher absorbance in visible bands is observed in flowering months except for NIR, which is correlated with other bands until June, but then shows higher reflectance in flowering months (Figure 4).

The NDVI profile of the meadow sites by elevation for two years (2017, 2018) show alignment with flowering months (Figure 4G,H). NDVI is most elevated in late July and August (Figure 4D,E); coincidentally, there is a ramp-up late June followed by a decline in greenness index after August (Figure A5). NDVI values between −0.1 to +0.1 signifies snow, which is evident in months until June (Figures A2 and A3). Additionally, NDVI variability is lower at the highest elevation, where vegetation is sparse, than at the lower elevations; also evident is the lag in peak NDVI by elevation (Figure A4A,B). We found that average NDVI was linked to flowering months July and August (Figure A4C), also, NDVI drops after plateauing in late July–early August (PS imagery, Figure A6).

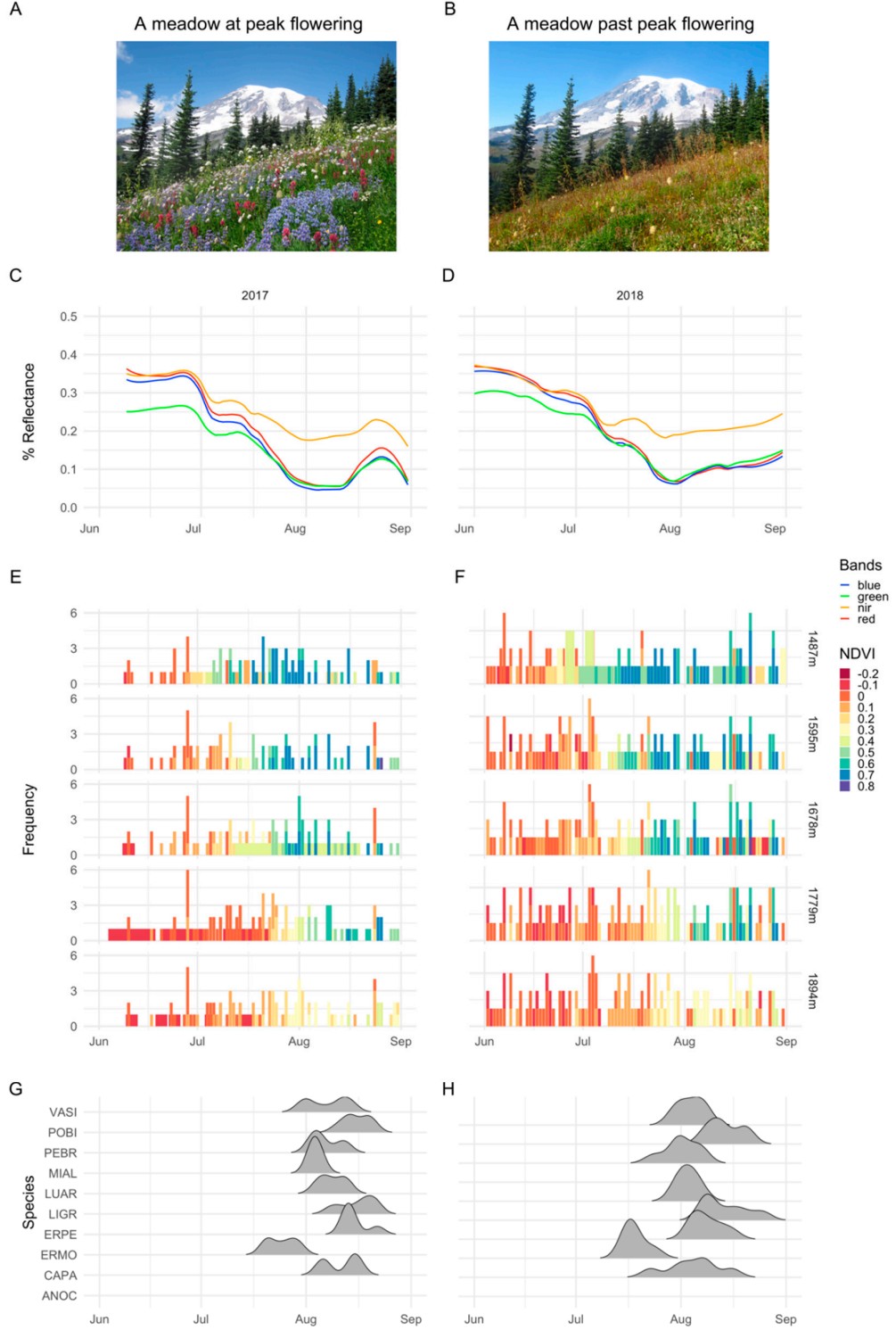

**Figure 4.** (**A,B**) A typical flowering meadow at peak, and a meadow past the peak flowering. (**C,D**) Reflectance profile of all the meadows across two years (2017 and 2018) using PlanetScope item PSScene4Band (type *analytic_sr*) in the Planet workflow (PS). (**E,F**) Normalized Difference Vegetation Index (NDVI) profile of all the meadow sites by elevation for two years (2017, 2018) using PS. On the alternate y-axis is the elevation in meters, and y-axis shows the number of PS captures/meadow having the corresponding NDVI threshold. The NDVI thresholds are determined by taking the average of NDVI metric across the entire meadow. (**G,H**) Flowering observations for years 2017 and 2018 from MeadoWatch; showing dominant 10 flowering species.

A large proportion of the variability in the spectral bands of PS was captured by the first two principal components (PCs) (Figure 5A,B). In peak flowering months (July and August), PC1 was positively correlated to mean NDVI, and negatively correlated to mean NIR reflectance values (Figure 5). Additionally, in peak flowering months, the reflectance in visible and NIR bands are negatively correlated to PC1. In non-flowering months, PC1 is positively correlated to reflectance in visible and NIR bands. Seasonal trends are evident, as snow starts declining in the meadows in May and continues with vegetative growth towards late June and flowering peaking around August. In typical flowering months July and August, NDVI values are strongly clustered with correspondingly higher positive mean values of NDVI in the range between 0.6 and 0.8 (Figure 5A,B). Spatial variability of flowering across elevations is also evident with lower elevations strongly tied to increased NDVI than higher elevations (Figure 5C).

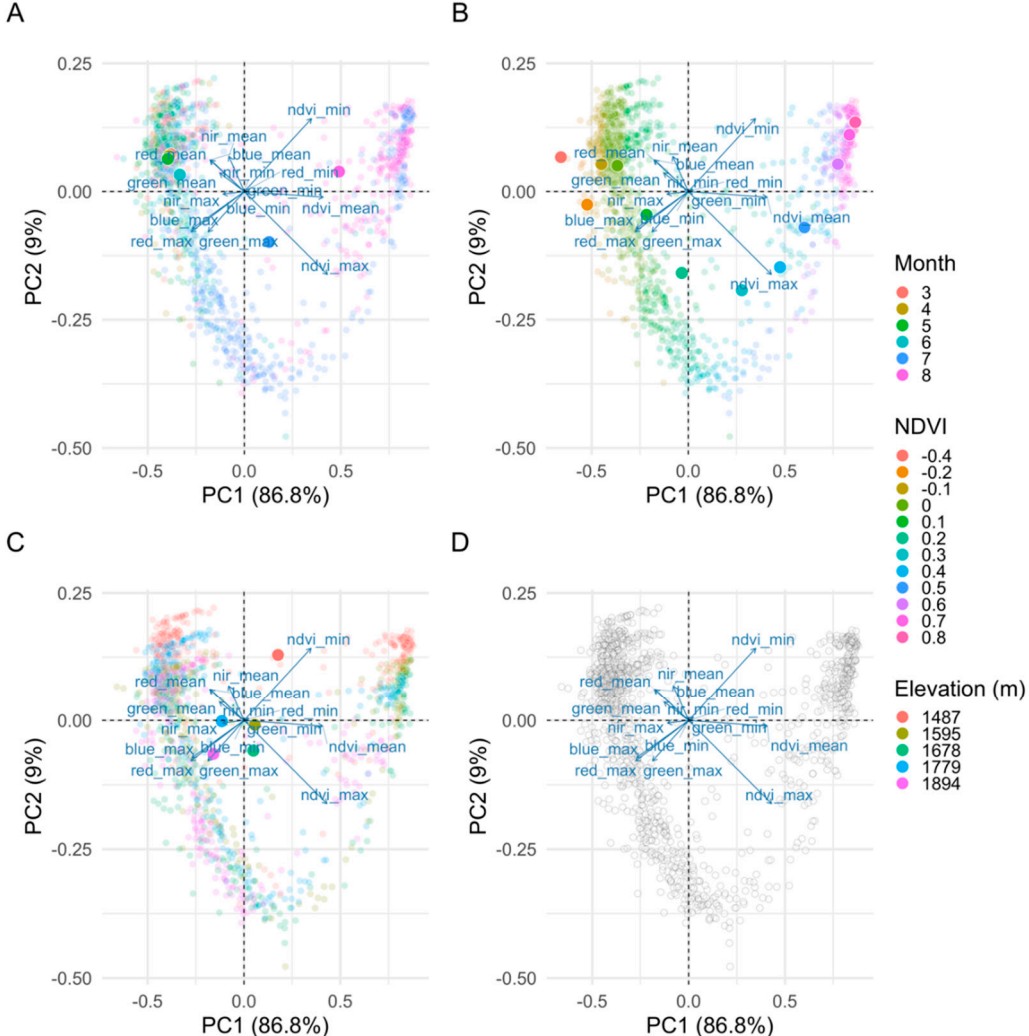

**Figure 5.** Biplot ordination from the principal component analysis (PCA) for 2 years (2017 and 2018) of spectral data from PS for the 5 meadow sites. (**A–C**) PC1 and PC2, overlain with the centroid (filled large dots) and average summary reflectance's captured in visible and near infrared (NIR) bands colored by month, NDVI, and elevation (blurry dots); (**D**) the ordination without any highlighting of the individual summary reflectance's. (**A**) Seasonal trends are evident; snowmelt in the meadows to flowering from upper left to upper right in the panel; blurry dots are colored by month. (**B**) Flowering months (late July and August) demonstrate higher positive mean values of NDVI in panel; blurry dots are colored by NDVI. (**C**) Flowering at sites showing the spread by elevation explained by increase in NDVI in the flowering months; blurry dots are colored by elevation of the meadow. (**D**) Strong correlation between visible bands, and NDVI metric that is orthogonal to NIR/visible bands.

## 3.2. Flowering Predictions Using Random Forest (RF)

Random Forest (RF) was trained to estimate the flowering window from PS and PS combined with coarser resolution satellite imagery. The results showed that the accuracy, i.e., correct classification rate when only PS was used was 70% (Kappa 0.25), when combined with coarser providers was 77% (Kappa 0.39). However, when only coarser imagery was used, accuracy was 72% (Kappa 0.37). PS explained the most variation at 55% and combined imagery without PS was the lowest with 29% (Table 2). We used a threshold of 0.25 to determine the predicted probability of flowering.

**Table 2.** Metrics when different types of imagery was used. Model combining the PS imagery with coarser providers yielded better results than PS only-based model.

| Metrics | PS + L8 + S2-1B | L8 + S2-1B | PS |
|---|---|---|---|
| Accuracy (%) | 77 | 72 | 70 |
| Median CV [1] RMSE | 0.29 | 0.27 | 0.31 |
| Median CV [1] Variation (%) | 50 | 29 | 55 |
| Kappa | 0.39 | 0.37 | 0.25 |

[1] 99 cross-validations with 0.10 proportion withheld at each run.

In our Random Forest (RF) variable importance analysis, NIR was less relevant than NDVI for identifying peak flowering. Mean values of reflectance in NDVI were most important in the RF model when only fine resolution imagery was used (Figure 6A), whereas, when using combined imagery, the blue band was most important (Figure 6B). However, NIR is integrated into NDVI as part of being a normalized measure with the red band, and NDVI does stand out in both the datasets, but NIR does not stand out as much as the red band.

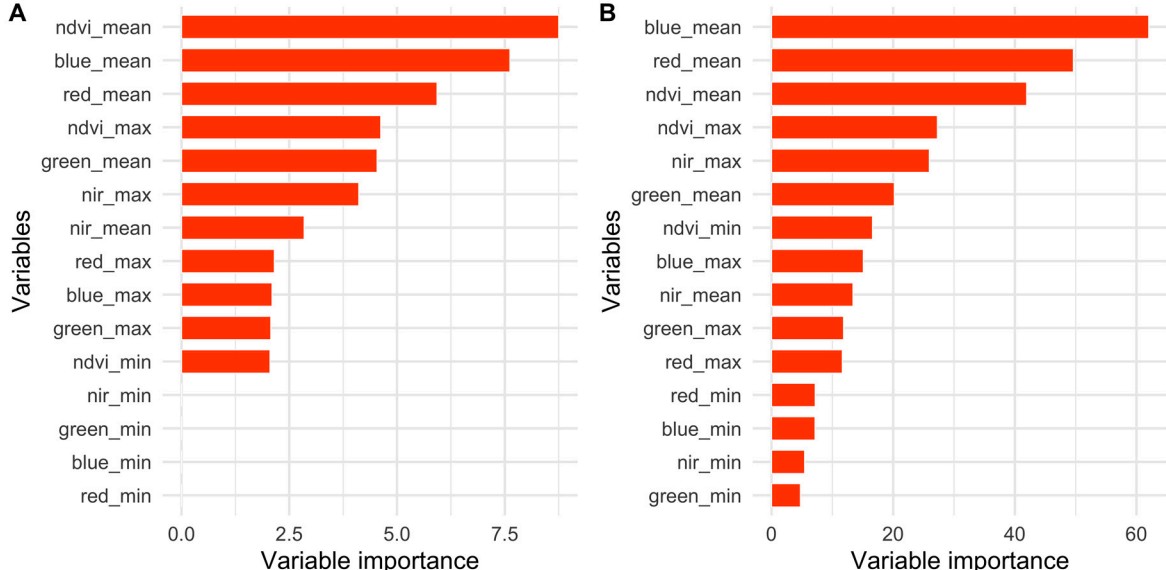

**Figure 6.** Relative importance of predictor spectral variables related to flowering when fine resolution imagery was used versus when fine-level imagery was combined with coarser resolution imagery. (**A**) Using PS only highlights NDVI as the top contributor. (**B**) PS, along with L8 and S2-1B, highlights the blue band as the top contributor.

Predicted peak phenology windows from spectral images aligned with observed phenology in some years than the others. The RF model captures the middle (or median) of the flowering window when compared to in-situ but tends to be longer, and the predicted window does not align in most cases when compared to MeadoWatch (Figure 7A,B). In comparison with both in-situ and MeadoWatch observations, there is over- and underestimation, which is evident in misaligned start of peak flowering

window in certain years/elevations and overlapping intervals in some years/elevations. For 2015, an exceptionally warm year, both Random Forest (RF) models predict with less overlap but exhibit longer flowering windows that show delayed onset of flowering. Both the models overpredict the start of the flowering in average years (2013 and 2014) than in a warm year (2015). The predicted flowering windows aligned better when combined imagery was used than when only the finer imagery was used.

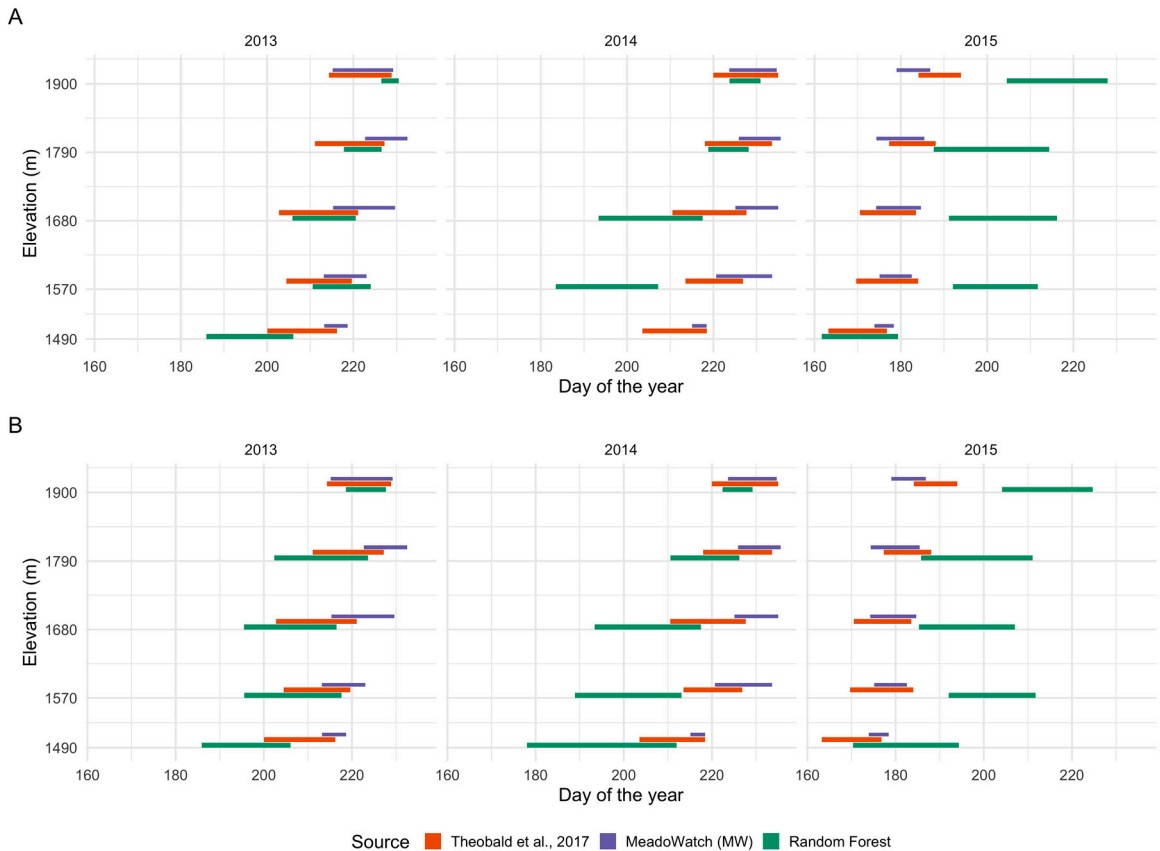

**Figure 7.** Qualitative comparison of observed window from the in-situ observations of (Theobald et al., 2017), MeadoWatch program, and Random Forest (RF) based flowering window. (**A**) Predicted and Observed peak flowering window when only 3-m (PS) resolution data was used for training. (**B**) Predicted and Observed peak flowering window when 3-m (PS) resolution, along with 10-m (S2-1B) and 30-m (L8) data, was used for training.

## 4. Discussion

Our study demonstrates that the timing of peak flowering in alpine meadows is detectable using fine resolution CubeSat imagery, and that the addition of coarser resolution imagery also substantively improves model accuracy. We also found that NDVI (a metric often used to quantify vegetative phenology) better predicted flowering phenology than did the NIR spectral band. The flowering window predicted from our model overlapped with the observed window in many site-year combinations but not in others, suggesting that accuracy is still an issue when using remote sensing imagery to detect flowering in meadows. Finally, more years where fine-resolution imagery overlaps with on-the-ground phenology data would likely have allowed for improved model and model assessment, as we only had 2 years of PS imagery to relate to our MeadoWatch observations.

We found that higher reflectance was observed in NIR and red bands when flowering occurred and can potentially be combined with reflectance captured in other bands for improved flowering detection. Meadow sites exhibited visibly higher reflectance in the NIR when flowering as compared to meadow green-up, an expected result as flowers decrease absorption after the initial green-up [15,30]. Strong absorbance was also found in red and green bands during flowering, as well equal absorbance

in the blue band when comparing to NIR. The blue band is typically associated with green-up time; it is a likely indicator of plant photosystems being active as allocation to vegetative growth is needed to support the energy required to produce flowers [31]. We found that NIR was correlated to flowering as is NDVI (which is a function of NIR). We also explored other indices, like green chromatic coordinate ($g_{cc}$) index, which is a proxy for greenness that has been found to be useful for detecting flowering other studies [32]. Our findings suggest that there is a decrease in greenness (i.e., $g_{cc}$) past peak green-up prior to flowering (Figure A5).

Use of visible and NIR bands are instrumental in phenological explorations but might be better when coupled with additional bands. We show that using visible bands (red, green, and blue) and NIR bands were sufficient in narrowing down the flowering window, but use of finer bands at the red-edge region might help further augment the flowering signal [33,34]. For example, use of SWIR has been found to be useful in accentuating signals of senescent vegetation. In addition, a recent announcement by Planet to provide additional bands (between 5 to 8) on next generation CubeSats could improve the phenological assessment at finer resolutions.

NDVI was found to be a significant predictor of peak flowering phenology in alpine/subalpine meadows. NDVI normalizes NIR and red bands and has been shown to be associated with green-up [35]. Furthermore, flowering must be preceded by peak green-up, which is evident by plateauing of NDVI (Figure A4A,B), signifying saturation because of chlorophyll accumulation [36]. Although reflectance in NIR visually shows the strongest pattern during peak flowering (Figure 4C,D), we actually found that a normalized measure like NDVI (which incorporates NIR) better predicts flowering (Figure 6A,B). The NDVI-based metric is usually applied in crop classification studies [37], but we also found it useful for detecting flowering in alpine meadows—an important result as it indicates this metric may also be applicable in studies which are looking to differentiate other phenological stages.

Despite the fact that NDVI was a reliable metric in our study, we recognize that it also has a number of shortcomings. Remotely sensed NDVI has been acknowledged as differing from true NDVI because of atmospheric effects and varying soil brightness, both of which can differ from image to image [38]. Although soil blotches are not common in alpine meadows except at very high elevations, the largest challenge is associated with canopy background limitations [39]. For this reason, Enhanced Vegetation Index (EVI) is often used as an alternative to NDVI to address the soil and atmospheric limitations [16]. We therefore suggest that EVI and the other related indices should be explored for their utility in predicting phenological stages beyond vegetative ones, including the combination of several indices to improve measurement accuracy.

We show that combining several types of satellite imagery leads to enhanced predictions of flowering phenology. Specifically, by combining 3-m spectral imagery with 10-m and 30-m imagery, our model predictions showed better overlap with observed flowering windows when compared to just the finer resolution model. However, the combined model tended to overpredict the start of the flowering windows (except in the anomalously warm year—2015). Similar studies have shown that multi-resolution data analysis improves results: "fuzzier" low resolution data can provide "big picture" information, and, when combined with the lower-resolution data, finer details can be revealed [40].

Our results also suggest that coarser 30-m Landsat imagery can be useful to infer peak flowering. The model parameterized with combined imagery (including finer scale Planet imagery) was able to infer peak flowering when applied to Landsat in years where only coarser scale imagery was available. Additional improvements to model predictions are possible by refining atmospheric corrections used for Landsat and Sentinel; in this study, we uniformly applied corrections across each individual band in order to account for atmospheric absorption, such as the effect of haze (Appendix A). However, these effects are not likely to be uniform. Future work should examine the effect of band-specific atmospheric corrections on phenology prediction accuracy.

Despite having an amalgam of satellite imagery of the meadows, one fundamental question is that are we able to distinguish the flowering signal from the background, i.e., soil, rocks or green vegetative growth [41,42]. PS comes with only 4 bands and has overlapping bands that might lead to pixel quality

issues [43,44], and other coarser providers have constraints of resolution and frequency of captures. However, meadow wildflower phenology progresses seasonally in a predictable manner. In the springtime, plants are covered with snow; when snow melts, they green-up, and then peak-flowering happens, and then snow falls again, covering all vegetation. This cycle culminates in about 4 months. We see this same 4-month cycle in the satellite signal through RGB composite and NDVI (Figure A6). Our method is bound by these constraints, but having more on-the-ground observations in time and even better satellite resolution (e.g., <3 m from DigitalGlobe) might be a significant next step. One option of combining imagery of various resolutions as we showed here is a promising approach. We plan on exploring the possibilities of augmenting the model training with more on-the-ground data and/or using the sub-meter imagery to create the training data.

Our work highlights the challenges to detect flowering from satellite imagery in heterogeneous plant communities, such as those found in alpine meadows. Previous studies reporting promising results were mostly conducted in simplified systems (e.g., monoculture crops) where there is greater uniformity in landscape features and phenological trajectories [16,45,46]. By contrast, the effects of elevation, vegetation composition, and heterogenous snow (in the early season) make it more challenging to predict flowering from satellite imagery in alpine meadows. In other montane systems, a modified approach that takes into account species level reflectance profiles might improve the flowering signal in mixed flowering species, e.g., use of vegetation color specific reflectance profiles in differentiating between flowering and green-up [15,47].

## 5. Conclusions

In this study, we tested whether fine-resolution imagery (<10 m) can detect flowering and whether combining multiple sources of imagery improves the detection process. By examining alpine wildflowers at Mt. Rainier National Park (MORA), we found that combined PlanetScope (from Planet Labs) data with coarser resolution but better quality imagery from Sentinel and Landsat satellites (10-m Sentinel and 30-m Landsat), resulted in higher accuracy in delineating the flowering season captured by ground-based phenological surveys than a finer 3-m resolution PlanetScope imagery.

Our methodology holds potential in quantifying climate-driven shifts in alpine meadows in support of both scientific and management goals. Our approach to wildflower phenology predictions using finer scale satellite imagery serve economic interests as Mt. Rainier National Park receives over 2 million yearly visitors, the majority of whom come in the summertime to enjoy the idyllic wildflower vistas. Better predictions of peak flowering could help park facilities to prepare for increased visitation and provide timely visitor updates.

**Author Contributions:** Conceptualization, A.J. and E.J.T.; methodology, A.J. and J.D.O.; software, A.J., J.O.; validation, A.J., and J.D.O.; formal analysis, A.J.; investigation, A.J.; resources, A.J.; data curation, A.J.; writing—original draft preparation, A.J.; writing—review and editing, A.J., J.O., E.J.T., J.D.O., A.T., and J.H.; visualization, A.J.; supervision, J.H. All authors have read and agreed to the published version of the manuscript.

**Funding:** AWS credits were provided by eScience Institute at University of Washington (UW) and UW-IT, and Microsoft Azure Cloud credits were provided through the Microsoft AI for Earth Grant to Aji John.

**Acknowledgments:** We acknowledge Planet's Ambassadors program support in providing the 3m PlanetScope imagery.

**Conflicts of Interest:** The authors declare no conflict of interest. The funders had no role in the design of the study; in the collection, analyses, or interpretation of data; in the writing of the manuscript, or in the decision to publish the results.

## Appendix A

Data refinement is spread out over reprojection and cropping; however, the bulk of the data processing is done after cropping. The data is first converted into Top-of-Atmosphere reflectance (TOA) and then to Bottom-of-Atmosphere reflectance (BOA) also known as surface reflectance (SR). This is because the data gathered comes as raw Digital Numbers (DN), and the DN values are simply scaled

values measured by the sensors and have no meaningful value. TOA is converted to BOA because BOA takes into account atmospheric effects, such as cloud cover, aerosol gases, etc. For Landsat, the equation for DN to TOA conversion is

$$\rho_\lambda = \frac{M_\rho \, Q_{cal} + \, A_\rho}{\cos(\theta_{SZ})}. \tag{A1}$$

In Equation (A1), $\rho_\lambda$ = TOA planetary reflectance, $M_\rho$ = band-specific multiplicative rescaling factor, $A_\rho$ = band-specific additive rescaling factor, $Q_{cal}$ = quantized and calibrated DN pixel value, and $\theta_{SZ}$ = solar zenith angle for solar correction.

$M_\rho$ is $2 \times 10^{-5}$, and $A_\rho$ is −0.1, which are both provided in the metadata file. The solar elevation ($\theta_{SE} = 90 - \theta_{SZ}$) angle is provided in the metadata file as an average of the entire tile, but Landsat provides a tool to get the $\theta_{SZ}$ for each pixel in a specific band. This is used instead of the approximation.

Once TOA reflectance is calculated, a scatter value is subtracted in order to get BOA reflectance. TOA is only reflecting from above the atmosphere. This scatter value is calculated by using a method called Frequency 50 Minus 0.008 (F50 0.8%) developed by GIS Ag Maps [48]. This is an image-based atmospheric correction model based on the Chavez Landsat TM histogram method [49]. This model only accounts for atmospheric scattering and assumes a constant haze value throughout the entire image. This constant haze value is determined by a relative scatter lookup table provided by GIS Ag Maps. This is by no means the most accurate way to derive surface reflectance as it is an image-based correction algorithm; however, it was used here because of its accuracy and simplicity.

**Appendix B**

Workflows for acquiring and analyzing Landsat, Sentinel, and Planet satellite images were written and executed on SWEEP [24], a scalable workflow management platform.

*Appendix B.1. Landsat*

The workflow begins with setting the input boundaries of the meadow sites, which are subsequently run against the USGS Earthdata Explorer API (EE API), along with the date ranges to fetch the available scenes. Scenes are available on EE API and AWS S3 (a cloud data storage product) as part of its open data program. EE API provides functionality to search and download Landsat imagery for free. The workflow was designed from the ground-up and is made available to ensure reproducibility. We chose to use both of these services to overcome an EE API limit on parallel downloads. Once a set of scenes is returned by EE API, the list of scenes was checked against AWS S3 for availability. Once matched scenes are found, they are downloaded from S3 and then re-projected (projection WGS 84 is used for this workflow), cropped, and radio-metrically corrected using the Landsat TM histogram method by (Chavez Jr, 1988). Next, statistical measures (min, max, and mean) are calculated across all the bands in visible and NIR spectra at the meadow level. Finally, the results are written to a comma-separated values (CSV) file which includes the statistical values for each band, the name of the feature (meadow site), and the date.

*Appendix B.2. Sentinel*

The workflow starts by setting a list of polygons referring to meadow sites. Next, the workflow gets the applicable AWS scenes using the *sentinelhub* Python package with a spatial and temporal filter. The results include the AWS S3 URL, as well as the date of each of the scenes, which are both extracted for all of the available scenes. Once the scenes are found for each meadow, retrieving and refining of the data for each scene is subsequently run in parallel. This includes fetching, re-projecting, cropping, and refining the image data. Re-projecting and cropping are similar to the Landsat implementation mentioned above.

The image data conversion for this workflow is less computationally demanding than that of the Landsat workflow because Sentinel-2 L1C data comes as Top-of-Atmosphere (TOA) reflectance. Thus, we only needed to subtract each band's scatter value to reach Bottom-of-Atmosphere (BOA) reflectance, also known as surface reflectance. The method used to find the scatter value is the same one used in the Landsat workflow, except that the Sentinel relative scatter table was used, as opposed to the Landsat relative scatter table. Both of the tables are provided by GIS Ag Maps [49] to find the scatter values for the bands.

*Appendix B.3. Planet*

Here, all the interactions are made via the Planet API through the SWEEP tasks. The workflow starts by using the meadow polygons to search for images within the desired date range. The results are image identifiers that are passed to the next task, which issues an API call to activate each image scene. The next task sends a clipping request to the API, whereby the meadow is clipped from the larger scene. A download request is then sent for each image, which, when ready, is furnished via a time-sensitive link. Finally, the images are downloaded, and metrics calculated for each band. The output is written to a CSV file that has band metrics time-stamped for each meadow.

## Appendix C

**Table A1.** Wavelengths (in nanometer, μm) of spectral bands by different satellite imagery providers.

| Band | Landsat 8 | Sentinel-2 | Planet |
|------|-----------|------------|--------|
| Blue | 0.45–0.51 | 0.45–0.52 | 0.45 = 0.51 |
| Green | 0.53–0.59 | 0.54–0.57 | 0.50–0.59 |
| Red | 0.64–0.67 | 0.65–0.68 | 0.59–0.67 |
| NIR | 0.85–0.88 | 0.78–0.90 | 0.78–0.86 |
| SWIR | 1.5–1.6 | 0.9–1.7 | N/A |

## Appendix D

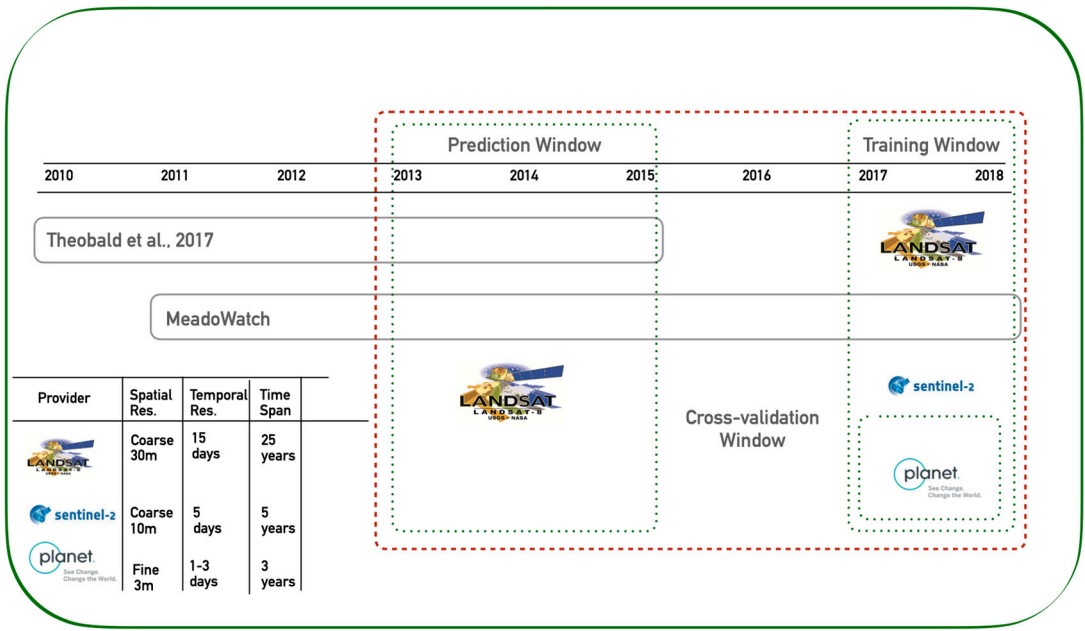

**Figure A1.** Time frame of fine resolution data versus coarse resolution data of the study.

## Appendix E

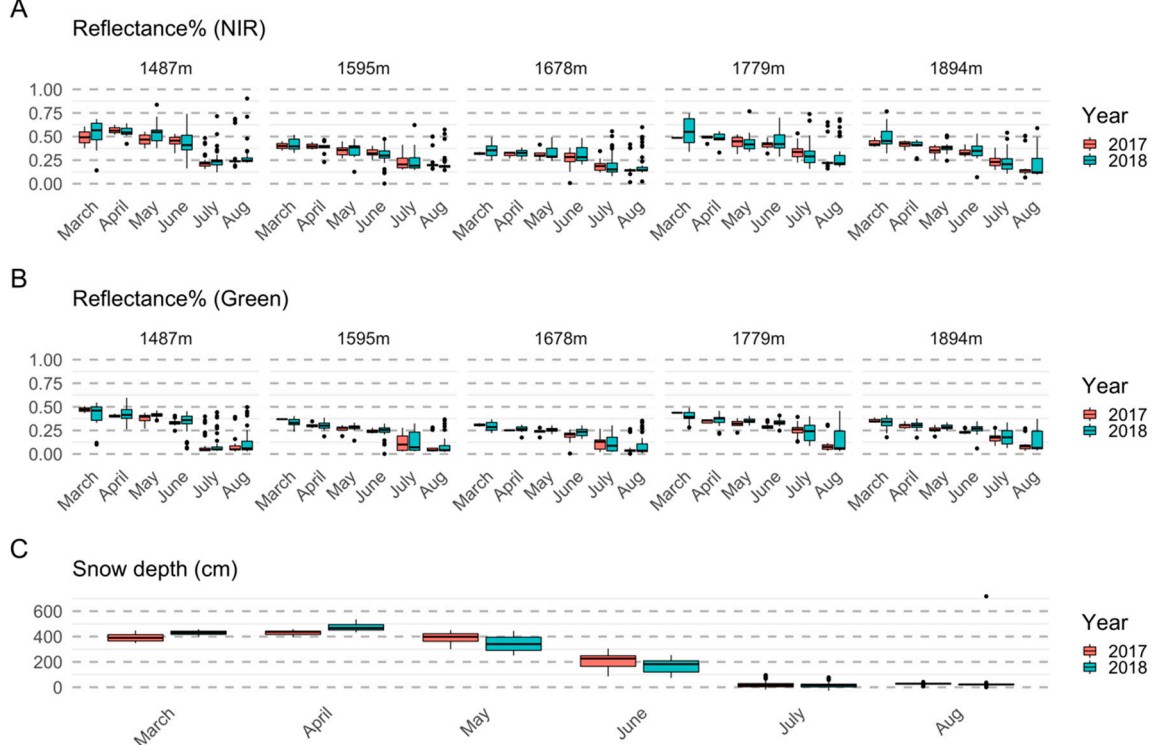

**Figure A2.** Planet reflectance comparison of NIR and green band showing snowmelt followed by green-up and flowering. (**A**) NIR reflectance is higher in snow months than in vegetative and flowering months, signified by positive correlation of reflectance's with snow-depth. (**B**) Green reflectance shows lower reflectance (i.e., higher absorption) after the snow-melt, signifying green-up. (**C**) Snow-depth plot from the metrological records highlighting snow-free months after June.

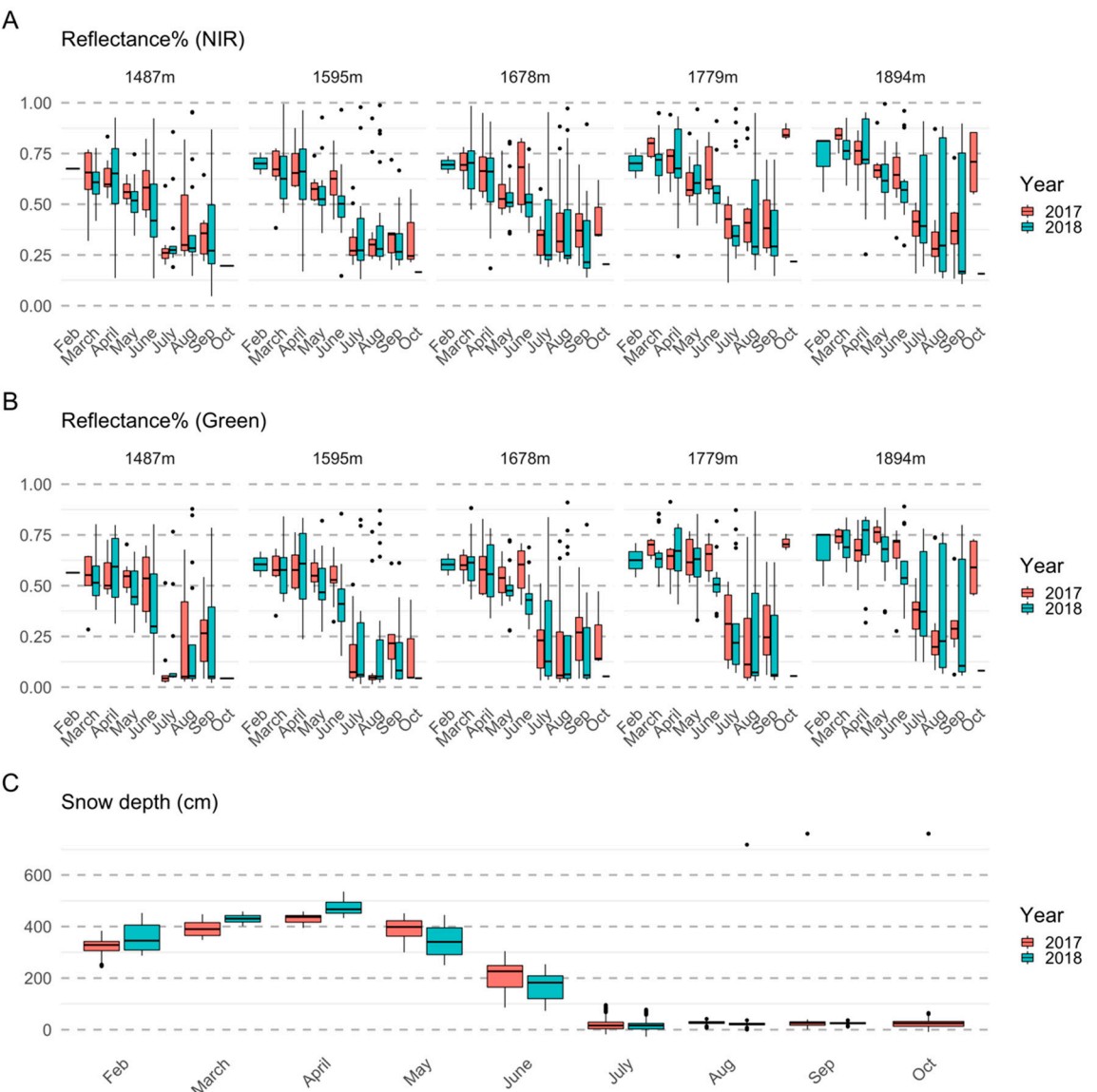

**Figure A3.** Combined Sentinel and Landsat reflectance comparison of NIR and green band showing snowmelt followed by green-up and flowering. (**A**) NIR reflectance is higher in snow months than in vegetative and flowering months, signified by positive correlation of reflectance's with snow-depth; however, note the upward trend of NIR in late September and October, signifying onset of snow or drying vegetation. (**B**) Green reflectance shows lower reflectance (i.e., higher absorption) after the snow-melt, signifying green-up. (**C**) Snow-depth plot from the metrological records highlighting snow-free months after June.

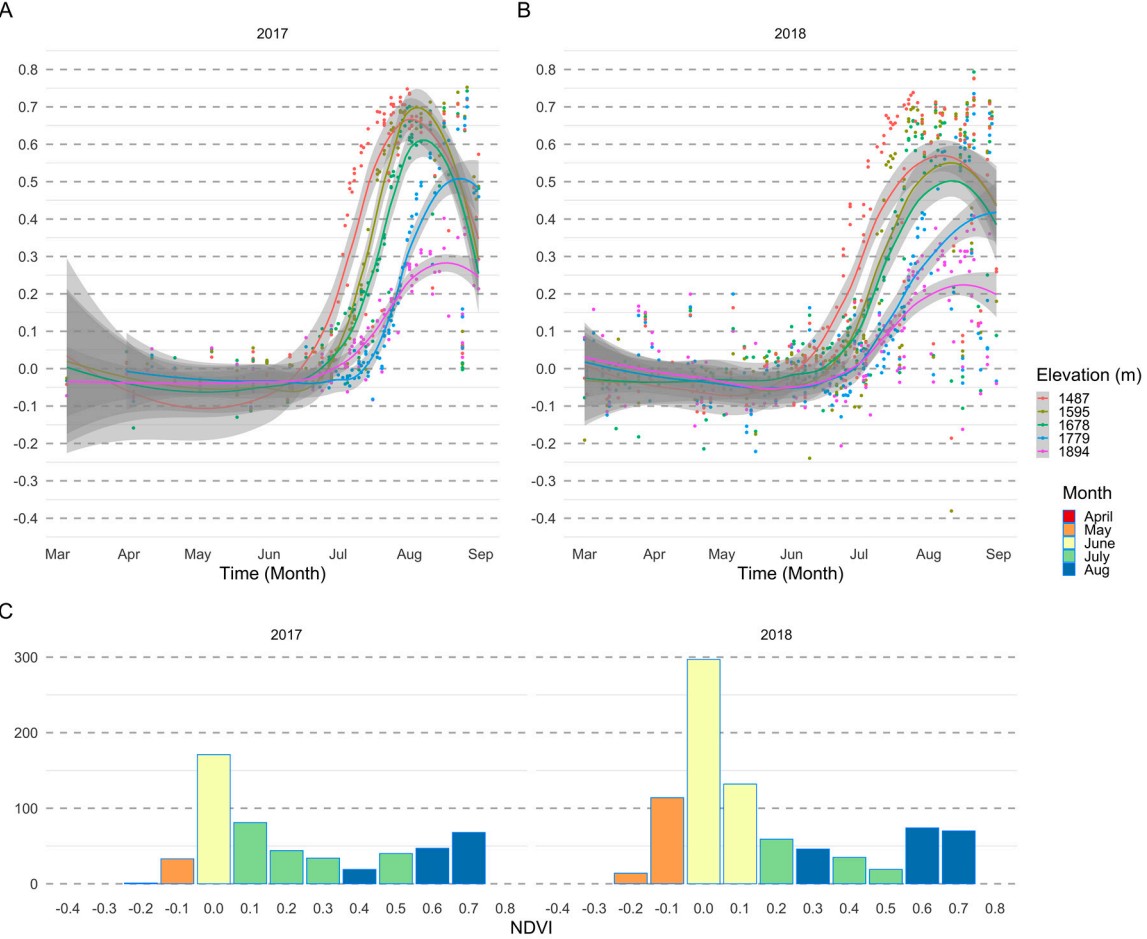

**Figure A4.** NDVI curve (loess smoothing) using Planet reflectance for years 2017–2018, showing the peak green-up around late July and also showing the phase lag by elevation. (**A**) 2017 NDVI values going from −0.2 to 0.8 capturing snow on the ground to flowering; also, notice the lag in peak NDVI confirming the snow-melt lag that is driven by elevation. (**B**) Similar patterns as 2017, 2018 NDVI values going from −0.2 to 0.8, capturing snow on the ground to flowering; also, notice the lag in peak NDVI confirming the snow-melt lag that is driven by elevation. (**C**) Histogram of NDVI values binned by year with the average month that it was observed in shaded.

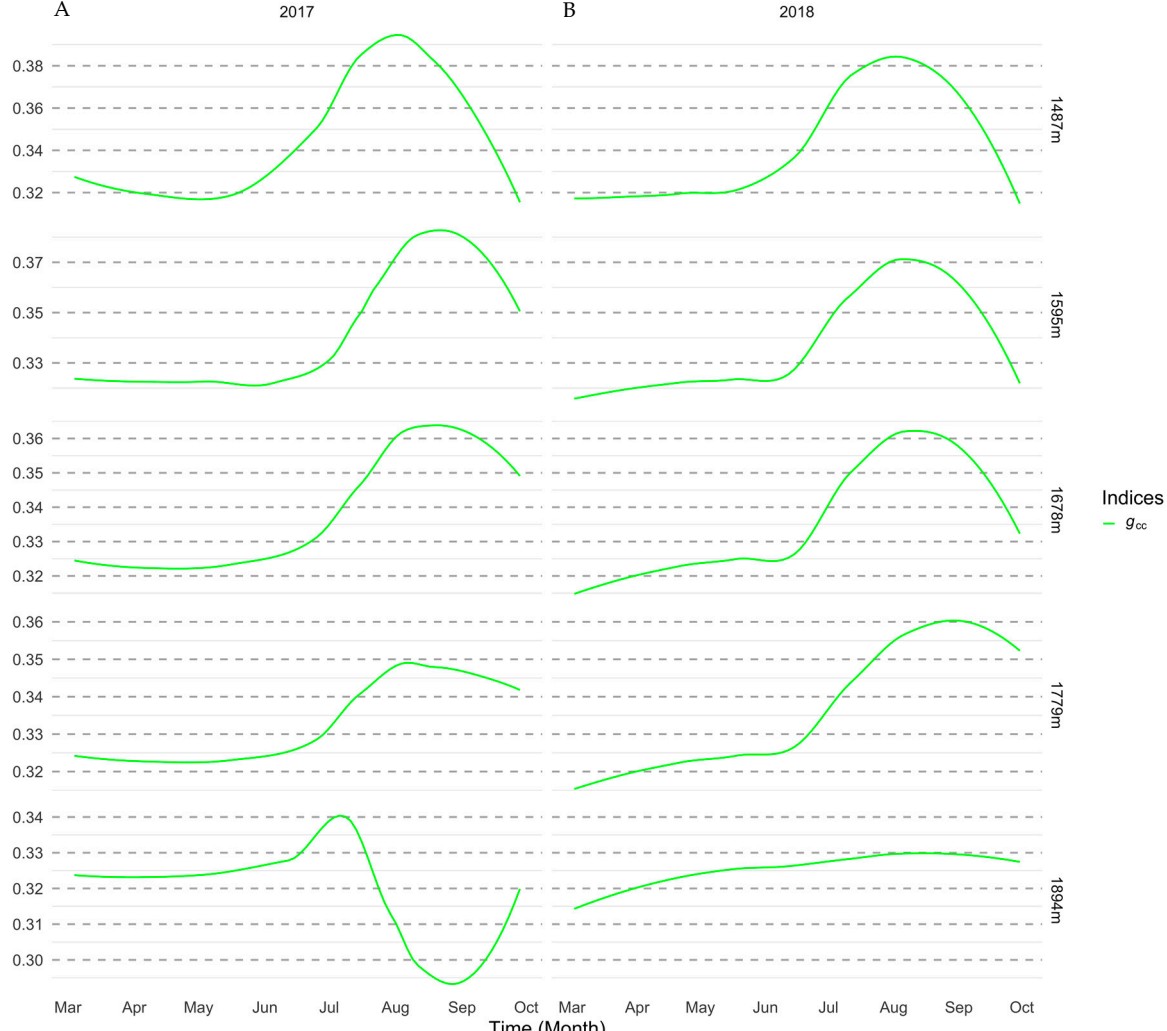

**Figure A5.** Green Chromatic Coordinate (*g_cc_*) that is a proxy of greenness calculated using Sentinel reflectance as data existed after August for years 2017–2018, showing the decline in green-up around late August but moderated by elevation. (**A**) 2017, capturing the ramp-up after late June right after snow-melt and decline after August; highest elevation sites have very short green-up. (**B**) Similar patterns as 2017, 2018 shows ramp-up in greenness around late June and drop in greenness after August.

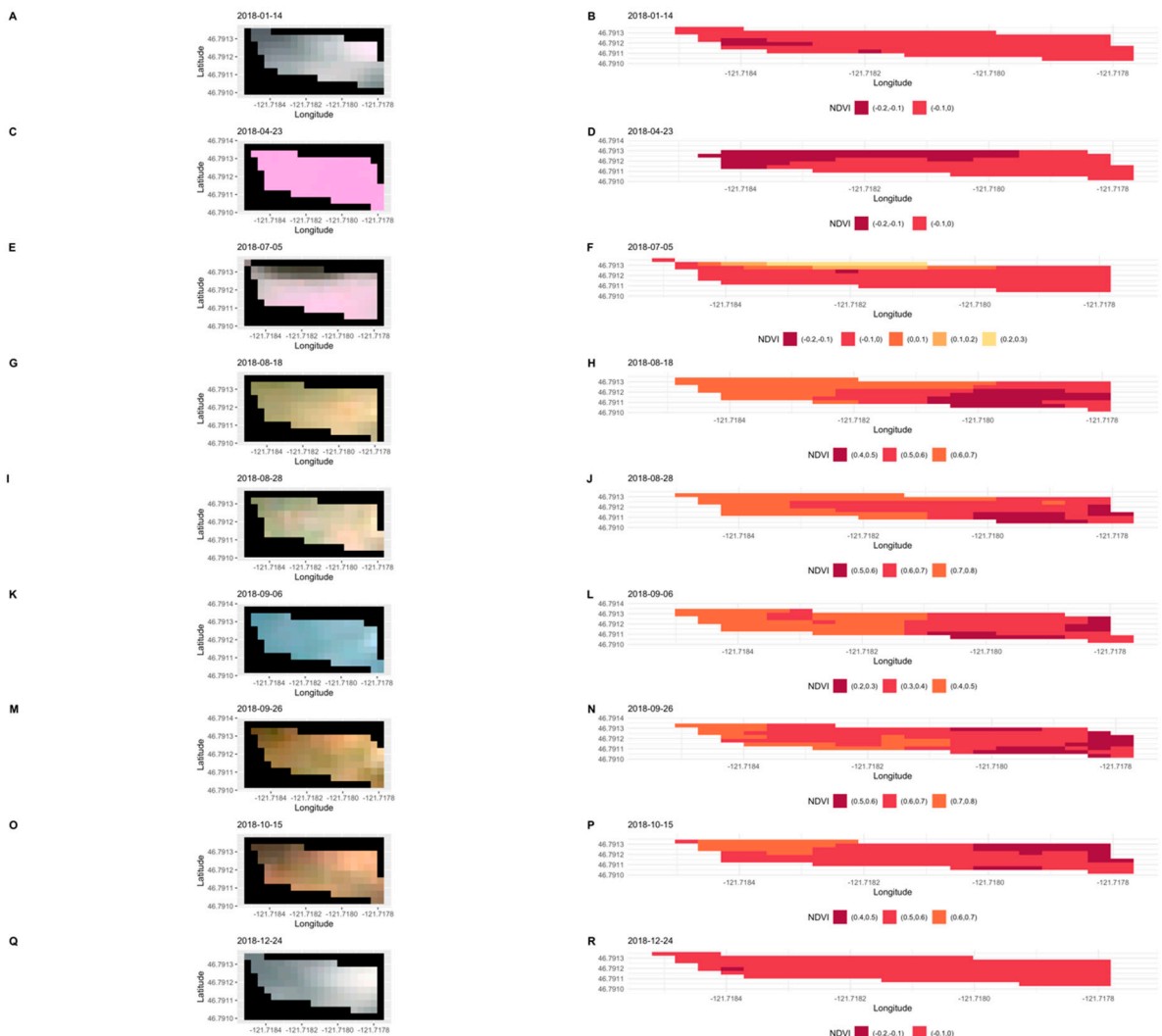

**Figure A6.** True color composite (red, green, and blue band combined) and corresponding NDVI of Meadow 4 (1779 m) for the year 2018. Column 1 for the panel gives a sense of how the meadow is changing during the whole year; starting from covered with snow (**A**,**C**,**E**), then meadow green-up and flowering (**G**,**I**,**K**), to senescence (**M**,**O**), and back to covered with snow (**Q**). Column 2 show the change in seasonality that is captured by NDVI; it shows in months till July NDVI remains below 0.2–0.3 (**B**,**D**,**F**), then a sharp increase in NDVI (**H**,**J**,**L**), and finally back to low levels of NDVI confirming snow (**N**,**P**,**R**).

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
