# Peer review of "Detecting Montane Flowering Phenology with CubeSat Imagery"

_remotesensing, doi:10.3390/rs12182894_

Round 1

Reviewer 1 Report

The paper was interesting. The study mapped wildflowers in the nine selected sites using multi-temporal satellite images and random forest. The overall design was easy to understand, but the manuscript can be better organized and presented. For example, I suggest separating the method section and add a workflow to overview each step. The result section also needs improvement to better support the discussion. Below is my detailed comments:

Line 57-59: Here should also talk about flowering mapping using Hyperspectral data and UAV.
Figure 1: How many Cubesat pixels in each polygon. Authors could add a table showing the area, elevation of each meadow.
Line 215-217: Why manually pick the threshold? Is the threshold applicable to a larger mapping area?
Line 226-229: Suggest put the paragraph into section 3.1
Figure 4 C and D: Frequency of what? Please clarify.
Figure 5: What's the difference between panel A B and C. Do they represent different years? What do large points represent? The PCA analysis shows the spectral variables related to the overall phenology, but how does this help the flowering classification?
Line 280-285: The feature importance of random forest could vary significantly. I suggest to run classification multiple times (e.g., ten fold) with randomly sampled training data and calculate the mean and STD of feature importance.

Author Response

Blue – Comments and Suggestions from the reviewer

Black – Our response

The paper was interesting. The study mapped wildflowers in the nine selected sites using multi-temporal satellite images and random forest. The overall design was easy to understand, but the manuscript can be better organized and presented. For example, I suggest separating the method section and add a workflow to overview each step. The result section also needs improvement to better support the discussion. Below is my detailed comments:

Thank you for the positive comments on our manuscript. We have addressed all the points raised by the reviewer and feel that these greatly improve the manuscript. We have focused on results by modifying Figure 4 and Figure 5 and elaborating our findings.

Line 57-59: Here should also talk about flowering mapping using Hyperspectral data and UAV.

Lines 59-60 Modified to indicate contribution of hyperspectral imagery.

Figure 1: How many Cubesat pixels in each polygon. Authors could add a table showing the area, elevation of each meadow.

Line 114 Per the suggestion, we have added a new table giving area and elevation of each meadow. For Cubesat one pixel is equivalent to a 3m by 3m area.

Line 215-217: Why manually pick the threshold? Is the threshold applicable to a larger mapping area?

Lines 228-230 We used true positive rate and true negative rate with ROC curve to get to the optimal threshold rather than manually picking. Threshold is probably applicable to larger mapping area as long as study system does not change, but it is an excellent point which we feel can be tested in follow up studies.

Line 226-229: Suggest put the paragraph into section 3.1

Lines 251-254 Makes absolute sense, we have moved it. Thank you

Figure 4 C and D: Frequency of what? Please clarify.

We have updated the caption for Figure 4C and D. Frequency refers to number of Planet images with a particular NDVI value by month.

Figure 5: What's the difference between panel A B and C. Do they represent different years? What do large points represent? The PCA analysis shows the spectral variables related to the overall phenology, but how does this help the flowering classification?

Lines 284-301 We echo the confusion in the PCA plot, we were intimated of the same by other reviewers as well. We have updated Figure 5 by adding a fourth panel just showing the ordination and have thoroughly revised the caption. Furthermore, we have augmented the plot to show the variables more clearly.

Panel A, B and C biplots are the same but are colored to show importance of month, NDVI and elevation in the ordination space. Large points refer to centroid variable being highlighted i.e. in A, its month, in B, its NDVI and in C, its elevation. PCA helped us to independently confirm the importance of spectral variables.

Line 280-285: The feature importance of random forest could vary significantly. I suggest to run classification multiple times (e.g., ten fold) with randomly sampled training data and calculate the mean and STD of feature importance.

Absolutely, and we recognize that, and in our case, single regression/classification trees can vary considerably as well, but random forests account for this by constructing thousands of trees with different subsets of observations and variables and reports the average of all trees. So, the method has accommodated for it already.

Reviewer 2 Report

Dear authors

The manuscript was well organised. I believe the readers of Remote Sensing will be interested in this study. I have some minor comments to improve the current manuscript. 

1: Fig.1: The scale of distance was wrong. 

2: Fig. 5: It's good to change the shape of coloured circle for "Month". For instance, "X". Coloured circles for "Month" and "NDVI" are very confusing. 

3:  Line 293-297: Really? I could not understand the relationship between Fig. 7 and this sentence. 

4: I think it's good to show the typical satellite images (e.g., RGB, NDVI) for each month. 

5: You used "inches" and "degree Fahrenheit". I believe we should use the SI unit. 

6: Please check and revise the style of references. 

Best wishes,

Author Response

Blue – Comments and Suggestions from the reviewer

Black – Our response

The manuscript was well organised. I believe the readers of Remote Sensing will be interested in this study. I have some minor comments to improve the current manuscript. 

Thank you for the positive comments on our manuscript. We have addressed all the points raised by the reviewer and feel that these greatly improve the manuscript.

1: Fig.1: The scale of distance was wrong. 

Revised the figure to show the correct scale. Also, modified Figure 1 to show the meadows and the MeadoWatch sites clearly.

2: Fig. 5: It's good to change the shape of coloured circle for "Month". For instance, "X". Coloured circles for "Month" and "NDVI" are very confusing. 

Lines 284-301 Thanks for this suggestion, we have updated Figure 5 by adding a fourth panel just showing the ordination and have thoroughly revised the caption. Furthermore, we have augmented the plot to show the variables more clearly.

3:  Line 293-297: Really? I could not understand the relationship between Fig. 7 and this sentence. 

Lines 346-355 revised in depth, and now reads “in some years than the others… finer imagery was used”.

4: I think it's good to show the typical satellite images (e.g., RGB, NDVI) for each month.

Lines 435-447 Thank you for the excellent suggestion, we have added a new supplemental figure (E5) showing both RGB composite and NDVI for one meadow for year 2018. The composite and NDVI at pixel scale capture the transition of the meadows. Additionally, we have touched on the constraints in earth observation studies like ours in discussion.

5: You used "inches" and "degree Fahrenheit". I believe we should use the SI unit. 

We revised supplemental figures E1 and E2 to show snow depth in cm.

6: Please check and revise the style of references. 

We re-applied the MDPI references style.

Reviewer 3 Report

The paper attempts to determine whether satellite data can accurately detect flowering dates in alpine environments. It is well structured and well written. Unfortunately, it’s not clear to me whether the method is really detecting a signal from the flowers, or whether the machine learning method is picking up a different signal from the satellite data.

Major issues

Whether satellite data is able to detect flowering species will depend to a large extent on the spatial coverage of the flowers during peak flowering. No data is presented to allow the reader to judge how much of a cubesat/Landsat pixel is likely to be covered by flowers during the peak flowering period. This is needed to determine whether random forest is really picking up a flowering signal, or whether it is peaking some other phenological signal.

There are various ways of dealing with this issue.

The simplest is to provide some data on typical fractional coverage of the flowers during the peak of the season and also include some photographs of the area in peak flower. This will enable readers to determine the likely coverage of the flowers in a satellite pixel and whether this will produce a strong enough signal in the satellite data to be detectable. Plus, adding a new section in the discussion about the fundamental constraint in EO that for something to be detectable it needs to have a  distinct spectral signature and it needs to cover enough of the pixel to produce a strong signal that is distinct from the background. This should cite references like: https://www.sciencedirect.com/science/article/pii/S0034425701002073?via%3Dihub

Finally, revising figure 4 to more clearly show the peak flowering season and the associated spectra (by displaying both on the same graph) would also help convince readers that the authors are actually detecting a flowering signal.

Minor issues

Line 84: The sentence starting on line 84 would be best split into 2 sentences.

Line 104: rephrase, so don’t refer to ‘long term study’s twice

Line 120: please (briefly) expand on what the quality issue was, if it’s a known issue (like the georeferncing/cloud-masking), then please include a reference e.g.

Coluzzi, R.; Imbrenda, V.; Lanfredi, M.; Simoniello, T. A first assessment of the Sentinel 2 Level 1-C cloud

mask product to support informed surface analyses. Remote Sens. Environ. 2018, 217, 426–443.

Yan, L.; et al., 2018, https://doi.org/10.1016/j.rse.2018.04.021

Line 123: I would prefer the n=2893 earlier in the sentence, but this is optional.

Line 130: add ‘analysis’ after ‘Landsat’.

Figure 2 caption: change caption so that it’s clear that this is ground data – some readers will skim through and just look at the figures.

Line 154: Was this done in this work, or is their a reference for this?

Line 155: replace ‘was a year where there is’ with ‘showed’

Line 177: rephrase the final line of this section, as I’m not sure what you mean. Are you referring to the fact that machine learning algorithms require comprehensive training data sets, which cover all the species that will be encountered. If so, this issue of requiring appropriate training data might be better covered in the discussion.

Line 191: rephrase ‘observed known flowering months’

Line 199: clarify whether this is 30m^2 or 30m x 30m

Line 226: Make it clear what you are referring to ‘The PS data showed…’

Line 228: should this be decrease, rather than increase

Figure 4 caption: Figure 4: I think that these figures would be clearer if they focused more on the growing season and less on the snow-covered part of the year.  June-September are the key months. Would the figures be clearer if they just covered these months?

Line 240: This section could be a bit clearer, as it jumps between the different wavelengths and different parts of the growing season. Line 241 about the blue repeats the discussion about the blue band in line 238.

Line 246: Figure 4C and 4D:I think this needs more explanation as i'm not clear what the frequency is. It probably just needs another sentence to say 'the y-axis shows the number of pixels/sites exceeding the threshold. The NDVI thresholds are determined by....'

Figure 5 caption:  The title needs to be clearer. It explains the key point that panels a-c make, but not what they are, so needs something along the lines of:

Panels a-c show PC1 and PC2, overlain with the study sites (filled dots) and pixel values (blurry dots). a) shows month, b_ shows NDVI and c) shows elevation.

The ordination axes titles are unclear. It might make sense to add a fourth panel, just showing the ordination results by themselves.

Line 272: I think it would be useful to add a sentence reminding the reader what RF is predicting.

e.g. 'Random Forest was trained to estimate the flowering window from PS and PS and coarser resolution satellite data. The results showed...'

Line 274: please check that values, they don’t seem to match the table.

Line 280: you say that NIR was less relevant, but not what it’s not less relevant than.

Figure 7 – Would an x-plot be better with observed and predicted doy for the start date. Followed by a second x-plot for percentage overlap between observed flowering and predicted flowering date.

The current text describing Figure 7 doesn't seem to summarise Figure 7 very well.

Line 308-311: This is a much better discussion of figure 7, than the text in the results section.

Line 315: change ‘combined reflectance’ to ‘combined with reflectance’

Author Response

Comments and Suggestions for Authors from Reviewer 3

Blue – Comments and Suggestions from the reviewer

Black – Our response

The paper attempts to determine whether satellite data can accurately detect flowering dates in alpine environments. It is well structured and well written. Unfortunately, it’s not clear to me whether the method is really detecting a signal from the flowers, or whether the machine learning method is picking up a different signal from the satellite data.

Thank you for the positive comments on our manuscript. We have addressed all the points raised by the reviewer and feel that these greatly improve the manuscript.

Major issues

Whether satellite data is able to detect flowering species will depend to a large extent on the spatial coverage of the flowers during peak flowering. No data is presented to allow the reader to judge how much of a cubesat/Landsat pixel is likely to be covered by flowers during the peak flowering period. This is needed to determine whether random forest is really picking up a flowering signal, or whether it is peaking some other phenological signal.

There are various ways of dealing with this issue.

The simplest is to provide some data on typical fractional coverage of the flowers during the peak of the season and also include some photographs of the area in peak flower. This will enable readers to determine the likely coverage of the flowers in a satellite pixel and whether this will produce a strong enough signal in the satellite data to be detectable. Plus, adding a new section in the discussion about the fundamental constraint in EO that for something to be detectable it needs to have a  distinct spectral signature and it needs to cover enough of the pixel to produce a strong signal that is distinct from the background. This should cite references like: https://www.sciencedirect.com/science/article/pii/S0034425701002073?via%3Dihub

Finally, revising figure 4 to more clearly show the peak flowering season and the associated spectra (by displaying both on the same graph) would also help convince readers that the authors are actually detecting a flowering signal.

Lines 463-475 We really appreciate this comment, the heart of remote sensing. We have modified the figure 4 to add panel A and B that shows a meadow in summer (peak) and in fall (post peak). We have also added a new supplemental figure (E5) showing both RGB composite and NDVI for one meadow for year 2018. The composite and NDVI at pixel scale capture the transition of the meadows. Additionally, per your suggestion, we have touched on the constraints in earth observation studies like ours in discussion to say, “Despite having …training data”.

Minor issues

Line 84: The sentence starting on line 84 would be best split into 2 sentences.

Line 85 We have rewritten into two sentences to indicate the origin of data sets and what it entailed.

Line 104: rephrase, so don’t refer to ‘long term study’s twice

Line 106 Edited the text to remove repetition.

Line 120: please (briefly) expand on what the quality issue was, if it’s a known issue (like the georeferncing/cloud-masking), then please include a reference e.g.

Coluzzi, R.; Imbrenda, V.; Lanfredi, M.; Simoniello, T. A first assessment of the Sentinel 2 Level 1-C cloud

mask product to support informed surface analyses. Remote Sens. Environ. 2018, 217, 426–443.

 Yan, L.; et al., 2018, https://doi.org/10.1016/j.rse.2018.04.021

Line 125 The issue was with the conversion of top of atmosphere reflectance to bottom of atmosphere reflectance for the meadows. For most of the dates we procured the imagery; we were getting insensible values. We did not investigate further on the root cause of erroneous reflectance values, but plan to investigate in further studies.

Line 123: I would prefer the n=2893 earlier in the sentence, but this is optional.

 Line 127 Amended, thank you.

Line 130: add ‘analysis’ after ‘Landsat’.

Line 137 Added, thank you.

Figure 2 caption: change caption so that it’s clear that this is ground data – some readers will skim through and just look at the figures.

Line 149 Good point, amended the caption to say its in-situ.

Line 154: Was this done in this work, or is their a reference for this?

Line 162-163 Yes, it was done for this work.

Line 155: replace ‘was a year where there is’ with ‘showed’

Line 163 Done, thank you.

Line 177: rephrase the final line of this section, as I’m not sure what you mean. Are you referring to the fact that machine learning algorithms require comprehensive training data sets, which cover all the species that will be encountered. If so, this issue of requiring appropriate training data might be better covered in the discussion.

Lines 188-192 We see the confusion, thank you for pointing it out. We meant that not observing a particular species or group of them at a particular time does not mean that it doesn’t exist in that meadow; this could be because of environmental conditions like late snowmelt, or warm temperatures. We have gone ahead to elaborate the point to say “assumes that 10 focal…assessment”.

Line 191: rephrase ‘observed known flowering months’

Line 204 Amended, thank you.

Line 199: clarify whether this is 30m^2 or 30m x 30m

Line 212 Amended to say 30m x 30m

Line 226: Make it clear what you are referring to ‘The PS data showed…’

Line 251 Good point, we have made the changes.

Line 228: should this be decrease, rather than increase

Line 253 Yes, we agree and have made the change. The rapid decrease in reflectance during July and August, precedes the bump in NIR reflectance in subsequent months.

Figure 4 caption: Figure 4: I think that these figures would be clearer if they focused more on the growing season and less on the snow-covered part of the year.  June-September are the key months. Would the figures be clearer if they just covered these months?

We really appreciate this comment; we had debated the same, we see that now that it has been echoed. We have modified the Figure 4 to show the metrics from June-September. Thank you.

Line 240: This section could be a bit clearer, as it jumps between the different wavelengths and different parts of the growing season. Line 241 about the blue repeats the discussion about the blue band in line 238.

Lines 256-269 We see the point; we removed the repetition and reorganized the paragraph to start with green-up and end with flowering.

Line 246: Figure 4C and 4D:I think this needs more explanation as i'm not clear what the frequency is. It probably just needs another sentence to say 'the y-axis shows the number of pixels/sites exceeding the threshold. The NDVI thresholds are determined by....'

Lines 283-301 We amended the caption for Figure 4C and D to clarify the meaning of Frequency. Thank you for the pointer.

Figure 5 caption:  The title needs to be clearer. It explains the key point that panels a-c make, but not what they are, so needs something along the lines of:

Panels a-c show PC1 and PC2, overlain with the study sites (filled dots) and pixel values (blurry dots). a) shows month, b_ shows NDVI and c) shows elevation.

The ordination axes titles are unclear. It might make sense to add a fourth panel, just showing the ordination results by themselves.

Figure 4. Thanks for this suggestion, we have added the fourth panel to only show the ordination and revised the caption. Furthermore, we have augmented the plot to show the variables more clearly.

Line 272: I think it would be useful to add a sentence reminding the reader what RF is predicting.

e.g. 'Random Forest was trained to estimate the flowering window from PS and PS and coarser resolution satellite data. The results showed...'

Lines 314-315 Good point, we have added a lead-in to remind the reader about RF predicting the flowering window. Thanks again for the pointer.

Line 274: please check that values, they don’t seem to match the table.

Lines 321-322 Fixed, thanks for double checking.

Line 280: you say that NIR was less relevant, but not what it’s not less relevant than.

Line 323 Amended the text to indicate that it is in comparison to NDVI

Figure 7 – Would an x-plot be better with observed and predicted doy for the start date. Followed by a second x-plot for percentage overlap between observed flowering and predicted flowering date.

The current text describing Figure 7 doesn't seem to summarise Figure 7 very well.

Line 308-311: This is a much better discussion of figure 7, than the text in the results section.

Lines 346-355 Thank you for pointing to lack in clarity, we have addressed it by delving into more detail on how well the Random Forest predicted windows fare well when compared with in-situ and MeadoWatch observations.

The suggestion to improve on the figure 7 was well received but we did not have the time to implement it.

Line 315: change ‘combined reflectance’ to ‘combined with reflectance’

Line 382 Done, thank you.

Round 2

Reviewer 1 Report

All comments are addressed.

Author Response

Thank you for reviewing the latest changes. We sincerely appreciate the past reviews. 

Reviewer 2 Report

Dear authors

Thank you for your revision. The manuscript was well revised. 

I have just one comment. 

In Figure 1, please check the scale in the right bottom. "2000 km" is strange. 

Best wishes,

Author Response

Thank you for reviewing the latest changes. We sincerely appreciate the past reviews. 

Your review 'In Figure 1, please check the scale in the right bottom. "2000 km" is strange.'

Thank you for your keen observation, we have updated the Figure 1, and have removed the scale for the world map. We decided to do so as it was not contributing to the overall figure. 

Reviewer 3 Report

The authors have satisfactorily addressed the comments raised in my previous review, so i am happy for the paper to be published in it's current form.

Author Response

(The authors gave the same response as above.)
